# Safetywashing: Do AI Safety Benchmarks Actually Measure Safety Progress?

**Richard Ren**[*1,2], **Steven Basart**[*1], **Adam Khoja**[1,3], **Alexander Pan**[3],

**Alice Gatti**[1], **Long Phan**[1], **Xuwang Yin**[1], **Mantas Mazeika**[1],

**Gabriel Mukobi**[4], **Ryan Hwang Kim**[1,5], **Stephen Fitz**[6], **Dan Hendrycks**[1]

[1]Center for AI Safety
[2]University of Pennsylvania
[3]University of California, Berkeley
[4]Stanford University
[5]Yale University
[6]Keio University

## Abstract

Performance on popular ML benchmarks is highly correlated with model scale, suggesting that most benchmarks tend to measure a similar underlying factor of general model capabilities. However, substantial research effort remains devoted to designing new benchmarks, many of which claim to measure novel phenomena. In the spirit of the Bitter Lesson, we leverage spectral analysis to measure an underlying capabilities component, the direction in benchmark-performance-space which explains most variation in model performance. In an extensive analysis of existing safety benchmarks, we find that variance in model performance on many safety benchmarks is largely explained by the capabilities component. In response, we argue that safety research should prioritize metrics which are not highly correlated with scale. Our work provides a lens to analyze both novel safety benchmarks and novel safety methods, which we hope will enable future work to make differential progress on safety.

## 1 Introduction

Benchmarks serve as crucial standards, providing metrics by which models and techniques are evaluated. The AI safety community has invested extensively in creating benchmarks aimed at measuring distinct safety-relevant properties [89, 60, 51, 12, 76, 28, 82]. While these benchmarks have driven significant advancements, there is a critical oversight: the performance on safety benchmarks intended to measure bias, ethics, adversarial robustness, or fairness is often strongly correlated with general capabilities benchmarks such as MMLU [29], MATH [30], and GSM8K [17]. This correlation means that simply enhancing the upstream general capabilities of models, such as by scaling parameters and increasing training data, often boosts performance across all benchmarks indiscriminately [41, 52].

This oversight is problematic because safety benchmarks have seldom been scrutinized for this correlation [103]. Consequently, this lack of scrutiny obscures the development of techniques that

---

[*]Equal Contribution.

38th Conference on Neural Information Processing Systems (NeurIPS 2024) Track on Datasets and Benchmarks.

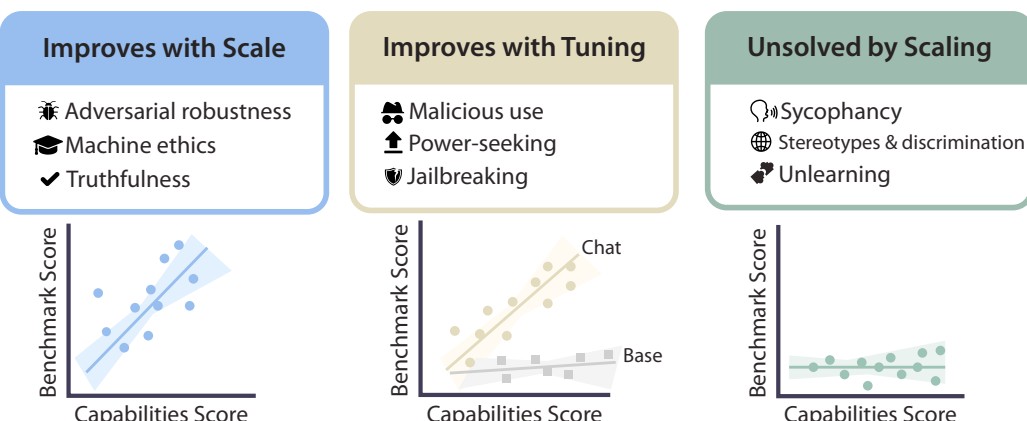

Figure 1: Our analysis identifies three classes of safety tasks according to the correlation between their scores and the capabilities scores. Tasks whose scores improve with scale have a positive correlation between benchmark scores and capabilities score. Tasks whose scores improve with tuning show a safer correlation on specific model classes, e.g., chat/instruct-tuned models. Finally, tasks whose scores do not improve naturally with model scale show no correlation between benchmark scores and capabilities scores.

specifically and differentially improve safety. Without clear and distinct metrics and goals, efforts to advance AI safety are hindered [97, 71]. The conflation of general capability improvements with safety-specific advancements not only misleads progress assessments but also undermines the incentive to develop targeted safety solutions [4]. To address this issue effectively, it is crucial to distinguish and prioritize safety-specific goals within the broader context of AI development.

Given this context, a pivotal question arises: how should the AI safety community allocate its efforts to differentially improve model safety? We can derive some insight from the "Bitter Lesson" [81], which observes that compute is becoming exponentially more available over time, and that AI research methodologies which optimize performance at a constant level of compute are subsumed by new paradigms that effectively leverage greater compute. Rather than over-indexing on the strengths and weaknesses of present-day models, this framework suggests that effective safety research should anticipate and address the flaws that will emerge or remain in future generations of models, and deemphasize issues likely to be resolved through general model scaling or the default trajectory of capabilities improvements.

Similarly, success in new safety methods should be measured not only by improvements in safety benchmark scores, but also by how much these methods make desired safety properties more correlated with scale. For example, Reinforcement Learning from Human Feedback (RLHF) [8, 58] has successfully associated toxicity reduction with model scale, an achievement that basic pretraining and instruction fine-tuning struggled to attain. By concentrating on properties and methods that specifically enhance safety independently of capabilities advancements, the safety community can make more effective use of its resources and significantly contribute to the development of safer AI systems.

## 2 Related Work

**Safety vs. capabilities.** One paradigm of measuring AI progress is a decomposition into datasets that measure "safety" vs datasets that measure "capabilities" [27]. While the distinction between safety and capabilities is sometimes blurred, safety research tends to study empirical phenomena that are negative side effects of model deployment [96, 66, 68, 74, 61], are malicious use of models [93, 105, 44], or do not improve with scale [11, 52]. In particular, a popular debate (e.g., between McKenzie et al. [52] and Wei et al. [94]) is whether a given safety dataset is in fact tightly correlated with scale. Our work addresses this debate through a meta-analysis of safety datasets, quantifying the degree to which safety datasets are entangled with capabilities.

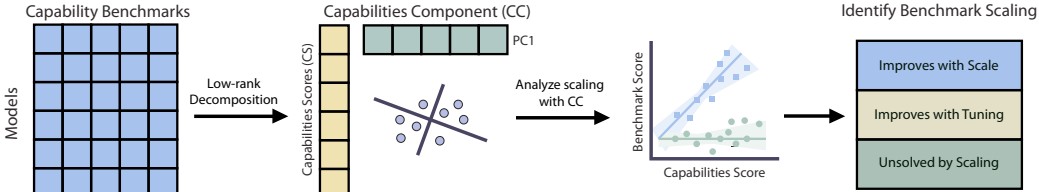

Figure 2: Illustration of the safety task identification pipeline. We first produce a matrix of scores for a set of language models evaluated on a set of capabilities benchmarks (first step). We extract the first principal component and use it to compute a capabilities score for each model (second step). We perform analysis of base and chat/instruct-tuned models on a variety of tasks representing major areas of AI safety (third step). Finally, we identify tasks whose scores are correlated with scale, tasks whose scores improve with scale only with chat/instruct-tuning, and tasks which are uncorrelated with scale (fourth step).

**Scaling laws and the Bitter Lesson.** The Bitter Lesson [81] argues that the main technique to improving ML models has consistently been scale. NLP has seen the biggest embrace of this trend, with developers focused on scaling the Transformer [88] with more data and compute [87, 85, 14, 1, 15, 83, 5, 6]. To aid in such engineering effort, there has been an extensive body of literature on quantitatively modeling scaling laws for loss, mapping out model performance as a function of compute and data [32, 41, 55, 33] or even hyperparameter choice [100]. Similar trends have taken hold in vision [24, 102, 26, 65] and robotics [86].

Scaling tends to improve not only training loss, but also downstream task performance [90]. A common finding is that models with lower pretraining loss also have higher accuracy on downstream tasks [95, 99, 35, 25, 23, 22, 19, 42]. Importantly, most prior work examines scaling laws from a model perspective (i.e., how does performance improve with scale), whereas our work examines scaling laws from a dataset perspective (i.e., how do benchmarks saturate with scale).

**Metrics and capabilities correlations.** Recent advances in observational scaling laws have provided a methodology that allows researchers to gain a deeper understanding of the underlying capabilities of machine learning models [41, 55]. Previous studies, such as those by [32, 15], have demonstrated that these scaling laws can predict model performance across various tasks. This body of work has established a foundation for using observational scaling laws as a powerful method for enhancing model training and evaluation processes by predicting performance trends based on scaling behavior. Recent research further supports the utility of these scaling laws, for extrapolating model performance, enabling a more nuanced assessment of model capabilities [75, 37].

However, while significant progress has been made in identifying and leveraging these underlying factors for general capabilities, there has been a noticeable gap in exploring how these scaling laws correlate with safety properties of deep learning systems. Although the identification of fundamental scaling relationships has been beneficial, there is a lack of research focusing on the implications of these relationships for safety datasets. Understanding how scaling impacts safety properties is crucial for developing datasets and benchmarks that can properly measure the intended effects and not by a "third variable" (i.e. capabilities). This paper aims to bridge this gap by examining the correlation between scaling laws and safety-specific characteristics, thereby providing insights that can guide the development of safer AI systems and future AI Safety datasets.

## 3 Capabilities Correlations for Evaluating Differential Progress on Safety

**Estimating capabilities using benchmark scores.** Inspired by prior work which applied factor analysis to matrices of model-benchmark scores [37], and concurrent to Ruan et al. [75], we apply spectral analysis of benchmark scores to identify a unified underlying *capabilities score* for models in terms of their performance on a range of benchmarks. Given a set of $n$ models and a suite of $m$ capabilities benchmarks (e.g. MMLU [29], Winogrande [77], GSM8K [17], etc.) we construct a matrix of scores $A \in \mathbb{R}^{n \times m}$, such that $A_{ij}$ is the score of the $i$-th model on the $j$-th benchmark, normalized so that columns have mean 0 and variance 1.

**Spectral analysis of capabilities scores.** Naive composite benchmarks usually weight their component tests equally, averaging test scores. A more principled approach can involve weighting component benchmarks according to the strength of their association with each other, with higher weight placed on benchmarks that account for greater variance in model performance across benchmarks. To achieve this, we compute a correlation matrix $C \in \mathbb{R}^{m \times m}$ associated with $A$, such that $C_{ab}$ is the correlation between task $a$ and task $b$ performance across all models. We extract the largest eigenvalue $\lambda$ of $C$ and its associated unit eigenvector $v$. The components of $v$ act as the weights of the composite benchmark, and $Av \in \mathbb{R}^n$ gives the capabilities scores of each model.

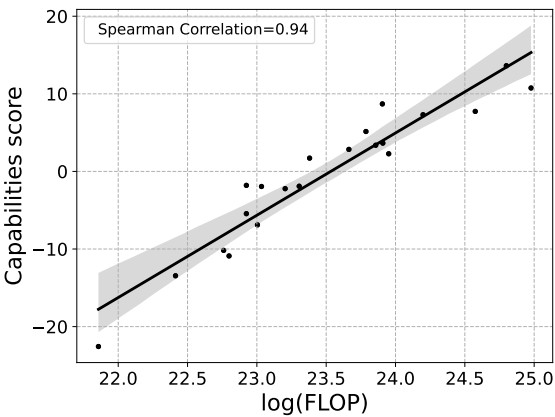

Figure 3: We observe a strong correlation between training FLOPs and relative capabilities score.

When $C$ is the Pearson correlation matrix $A^T A$ [64], $\sqrt{\lambda}$ is the largest singular value of $A$, and $v$ is its associated top principal component [21]. $\lambda/m$ then represents the proportion of total variance in normalized model scores explained by the principal component vector. Additionally, the outer product of the capabilities scores and benchmark weights $(Av)v^T$ is the best rank-1 approximation of $A$ [20]. However, using Pearson correlation can be sensitive to outliers, which becomes relevant when dealing with large model sets and a heterogeneous collection of benchmarks. For that reason, our analysis takes $C$ to be the Spearman correlation matrix [79], in which case $\lambda/m$ represents the explained variance in rank scores.

**Using capabilities scores to measure capabilities correlations.** To evaluate the relationship between a new benchmark and general capabilities, which we call the *capabilities correlation* of the benchmark, we can evaluate a set of models with known capability scores on the new benchmark and measure the correlation between capability scores and benchmark scores (we use Spearman correlation for these calculations as well). These general ability components allow for quantitative, intuitive, and principled evaluations of task relationship to general model abilities. Ultimately, however, these correlations depend on the set of models used, as well as the benchmarks chosen to produce their capabilities scores. In the Appendix, we perform a sensitivity analysis to explore the robustness of this methodology to different choices of models and benchmarks.

**Safety techniques can alter capabilities correlations.** In our analysis, we categorize models into distinct classes—base models and instruct (including chat) models—to better understand how different training paradigms impact performance on safety tasks. Base models, RLHF'ed models, light adversarial training, and future safety techniques could all be considered different model classes, with different profiles of capabilities correlations. Ideally, we should develop training regimens which produce high capabilities correlations with all relevant safety properties. By running separate analyses for each model class, we can identify the relative strengths of these techniques as models scale.

## 4   Results

We come to our central question: which tasks or datasets are correlated with capabilities? Towards answering this question, we analyze the overall capabilities scores as captured by tasks in 4.1, the tasks of Adversarial Robustness in 4.2, Bias and Toxicity in 4.3, Machine Ethics in 4.4, Malicious Use in 4.5, and Rogue AI Risk in 4.6.

To provide ease of understanding, we define a positive capabilities coefficient as yielding a safer system with scale, while a negative capabilities coefficient indicates less safe systems with scale.

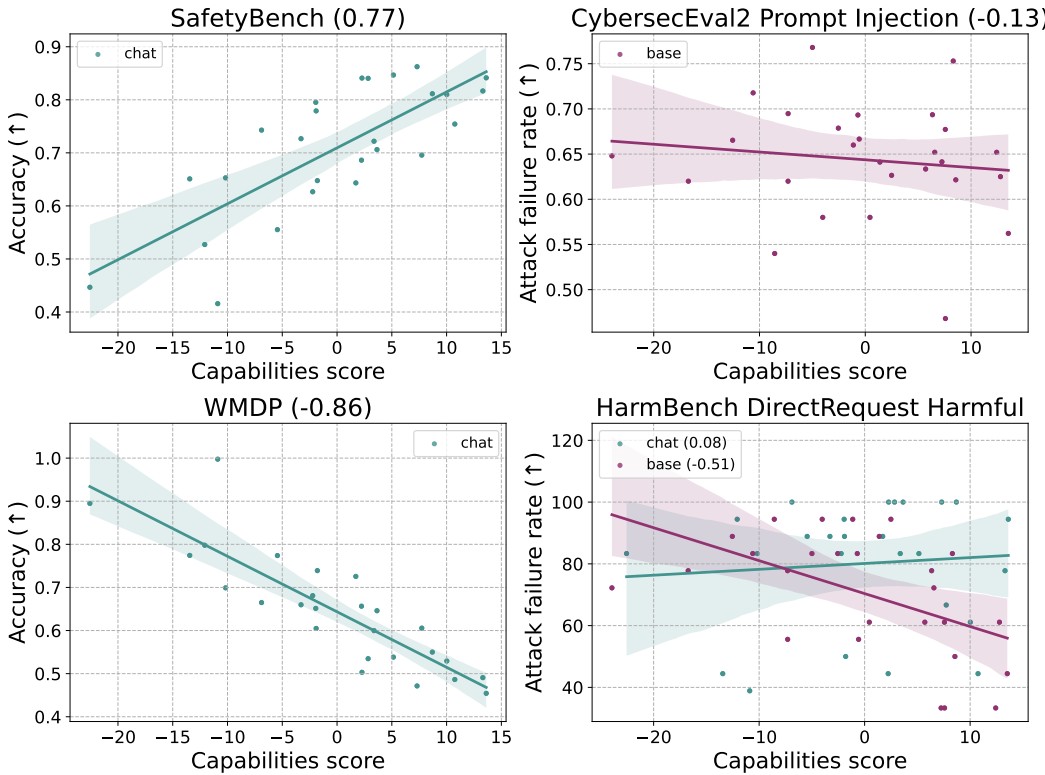

Figure 4: Observed correlations between capabilities scores and models' performance. Top left: safety task positively correlated with capabilities score. Top right: safety task not correlated with capabilities score. Bottom left: safety task negatively correlated with capabilities score. Bottom right: safety task where chat models are not correlated with capabilities score while base models are negatively correlated. Parentheses include the capabilities correlation of the corresponding benchmark.

## 4.1 General Capabilities and Overview of Model Class Correlations

**Most variance in capabilities datasets is explained by a capabilities component.** We run analyses for base and chat models, finding that 72% and 71% of variance is captured by the capabilities component respectively. We calculate the capabilities component from the following benchmarks: LogiQA [46], PIQA [13], Hellaswag [101], Winogrande [77], COPA [73], MedQA [40], ARC Challenge [16], MMLU [29], MATH [30], LAMBADA [62], Wikitext [53], GSM8K [17], GPQA [72], and BBH [9]. We also use a diverse set of model classes and derivatives to ensure robustness in results, as results can be skewed if they come from a derivative of one model (e.g. Llama-2 [87]); we list the 24 base models and 22 instruct/chat used for our analysis in the Appendix.

**The capabilities component is strongly correlated with scale.** We quantify the correlation of model capabilities scores with log FLOP for base (r=0.96) and chat (r=0.96) models, and plot chat models n in Figure 5. We calculate training FLOP via the approximation of 6 * params * train_tokens described in [41].

**Observed properties of capabilities correlations.** In our experiments, we observe three high-level categories of result:

a.) Some safety benchmarks [103] are already highly correlated with capabilities, obeying "scaling laws" (top left).

b.) Some safety benchmarks are not aligned with scale (top right) or are negatively correlated with scale (bottom left), obtaining worse safety properties as capabilities increase. At times, these problems are not solved by any type of model class.

c.) Some correlations are strengthened or weakened through safety techniques; for example, chat models exhibit on a higher correlation on CybersecEval2 MITRE [12], a task for measuring refusal to assist in malicious cyberattacks, than base models (bottom right).

In Figure 4, we examples of these scenarios. In the following sections, we continue to explore how instruction tuning affects models and highlight the need for alternative directions to be pursued across safety areas.

## 4.2 Adversarial Robustness

Adversarial robustness evaluates models' ability to maintain performance when faced with adversarial examples. In the vision domain, adversarial robustness is known to have different properties from general capabilities [92, 104]. However, the relation between general capabilities and adversarial robustness is less clear for LLMs. Many different adversarial robustness benchmarks have been developed to assess different aspects of their robustness [91, 77]. We now analyze whether these benchmarks measure novel properties or are highly correlated with general capabilities.

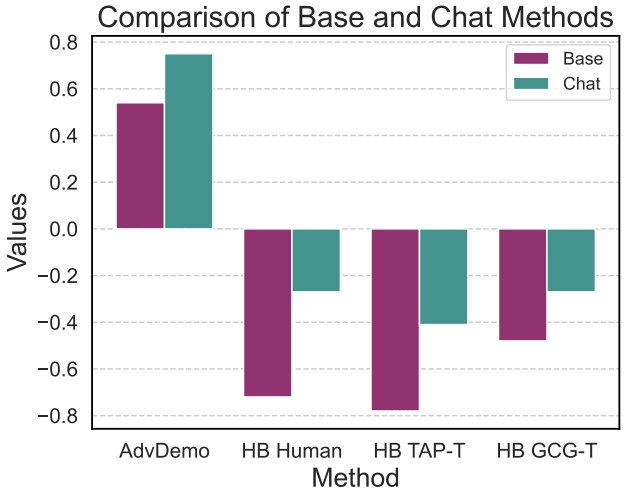

Figure 5: For many of the benchmarks we evaluate, capabilities correlations are higher (or less negative) among Chat models. This demonstrates that evaluating correlations for multiple model classes is crucial for understanding whether a benchmark will be solved as general capabilities improve.

We compute the correlation between the capability score and safety scores on the following benchmarks: AdvGLUE [91], AdvGLUE++ [34], AdvDemonstration [34], and Harm-Bench [51]. Full results on all datasets and model classes are in the Appendix.

**Some robustness benchmarks are correlated with general capabilities.** We find that some adversarial robustness benchmarks are moderately correlated with general capabilities, while others have low or even negative correlation. For example, AdvGLUE, AdvGLUE++, and AdvDemonstration have respective capabilities correlations of $0.68$, $0.58$, and $0.75$ for the instruct/chat model class. On the other hand, general capabilities are anti-correlated with robustness on HarmBench. The dynamic adversarial robustness benchmarks tested tend to have lower correlation, static adversarial benchmarks tend to have higher correlation. In other words, some robustness properties are likely to be solved as general capabilities improve, while others are not yet strongly correlated with capabilities.

**Different model classes have different scaling properties.** Just as adversarial training significantly alters the robustness properties of vision models, different classes of general-purpose AI models can yield different scaling properties for safety benchmarks. In Figure 5, we show how some LLM adversarial robustness benchmarks have higher capabilities correlations when using instruct/chat models. This demonstrates that improving the capabilities correlation of a safety benchmark is possible. Once the correlation reaches a high enough value, additional work on the benchmark is unnecessary, as it will be solved automatically as general capabilities improve.

## 4.3 Bias

We investigate bias datasets aimed at quantifying language models' propagation of social stereotypes and harmful preconceptions. It is well-known that pretraining on internet data introduces bias, and one might expect that training larger models on more data would increase the amount of bias present. We test this hypothesis by measuring the capabilities coefficient of different LLM bias benchmarks.

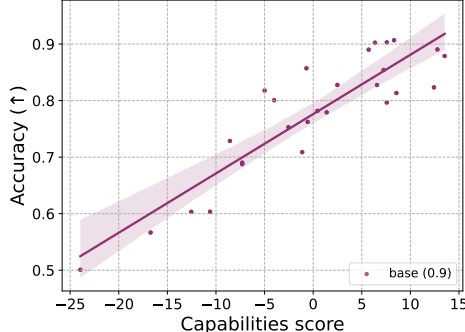

| Bias Evaluation | Capabilities Correlation | |
| --- | --- | --- |
| | **Base** | **Instruct/Chat** |
| CrowS-Pairs | -0.32 | 0.18 |
| Anthropic Discrim. | 0.36 | 0.40 |
| BBQ Ambig. | 0.25 | 0.29 |

Figure 6: Left: We find that for several common bias benchmarks, bias is not reduced by general capabilities improvements, indicated by low capabilities correlations. Right: However, on BBQ Disambiguated capabilities score is strongly correlated with reducing bias.

**Bias is often weakly correlated with capabilities, but not always.** Our findings reveal that for some bias measures, the capabilities correlation is weak as expected. For example, in Figure 6 (left) we show that BBQ Ambiguated [63], Anthropic Discrimination Evaluation [82], and CrowS-Pairs English [56] display this pattern across both base and instruct/chat models.

However, for other measures, improvements to general capabilities can actually reduce bias. In Figure 6 (right), we plot the capabilities score against accuracy on BBQ Disambiguated [63] and find that bias reduction is highly correlated with general capabilities. This observation contrasts with conventional wisdom, which suggests that scaling up models exacerbates bias due to associations in the training data [10].

## 4.4 Machine Ethics

Machine ethics benchmarks probe models' understanding of moral concepts. There are several benchmarks that analyze machine ethics, such as ETHICS [28] and STEER Rationality [70]. We report the capabilities correlation of these benchmarks in Table 1.

**High capabilities correlation.** We find that machine ethics benchmarks tend to be highly correlated with general capabilities. Many subsets of ETHICS have an extremely high capabilities coefficient for both base and instruct/chat models. These findings corroborate isolated observations of scale improving performance on machine ethics benchmarks [41, 50], indicating that internet-scale pretraining imbues LLMs with an understanding of ethics and morality. However, our results also show that this correlation is not identical across all areas of machine ethics. Some topics improve much more slowly with general capabilities, suggesting a need to ensure a balanced understanding of different ethical perspectives is present in models.

Table 1: Capabilities correlations for various machine ethics datasets. For brevity, we show instruct/chat models only, although correlations are also high for base models.

| Ethics Evaluation | Capabilities Correlation |
| --- | --- |
| ETHICS (Average) | 0.80 |
| ETHICS Commonsense | 0.72 |
| ETHICS Deontology | 0.41 |
| ETHICS Justice | 0.49 |
| ETHICS Utilitarianism | 0.74 |
| ETHICS Virtue | 0.77 |
| STEER Rationality | 0.54 |

## 4.5 Malicious Use

Malicious use evaluations test whether models can resist being exploited for harmful ends, including spreading misinformation or enabling cybercrime. Benchmarks like HarmBench [51], CyberSecEval2 [12], and WMDP [44] are used to assess the susceptibility of models to malicious use. To bypass refusal training, many of these evaluations also employ adversarial prompts. We analyze the capabilities coefficients of resistance to malicious use benchmarks under current models and present results in Table 2.

| Rogue AI Evaluation | Capabilities Correlation |
|---|---|
| MACHIAVELLI Power | 0.46 |
| MACHIAVELLI Utility | 0.48 |
| MACHIAVELLI Violations | 0.55 |
| Sycophancy | -0.73 |
| TruthfulQA MC1 | 0.83 |

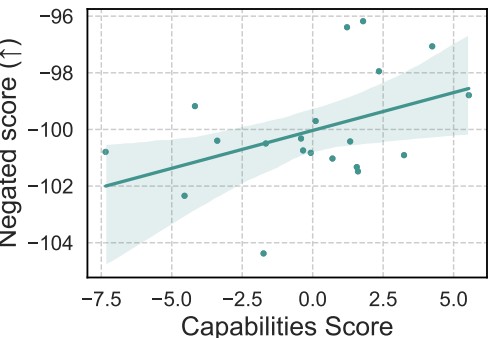

Figure 7: Left: Capabilities correlations on instruct/chat models on Rogue AI evaluations. Right: Capabilities correlations on instruct/chat models with accuracy on MACHIAVELLI Power.

**General capabilities exacerbate malicious use.** Many base models cause more harmful responses as their capabilities increase, as indicated by negative capabilities correlations. This includes many splits of HarmBench and CyberSecEval2, as well as WMDP (an unlearning dataset that penalizes high performance).

We find that instruction tuning weakens many capabilities correlations, indicating that models no longer become less safe with scale. In the MITRE task of CyberSecEval2, which measures refusal to participate in cyberattacks, the effect is even stronger, with the capabilities correlation changing from negative to positive.

These results demonstrate that instruct/chat models have improved over base models in their ability to leverage general capabilities to reduce malicious use risk. However, in most cases the correlations remain negative or weak, suggesting there is still considerable work to be done on this problem.

## 4.6 Rogue AI

Rogue AI risk evaluations probe risks related to deceptive model behavior, dishonesty, and power-seeking tendencies. Previously, it was unknown whether models become more power-seeking as they scale. We report the capabilities correlations of these benchmarks in Figure 7 (left).

**Power-seeking tendencies decrease with scale, but sycophancy does not.** On the MACHIAVELLI dataset [60], we find that measures of power-seeking tendencies and ethical violations decrease as general capabilities improve, with moderate capabilities correlations ranging from $0.46$ to $0.55$. On the other hand, sycophancy [67] becomes worse as models become more capable, with a capabilities correlation of $-0.73$. This highlights how different aspects of rogue AI risk are correlated with general capabilities to different extents.

Unlike power-seeking and sycophancy, we find that TruthfulQA MC1 variance is strongly correlated with general capabilities. This could be explained by training leakages or may indicate that models are able to discern fact from human falsehood as capabilities advance. Regardless, we find that TruthfulQA does not seem to measure a meaningfully different metric from capabilities benchmarks.

Table 2: Malicious Use Evaluations and Metrics

| Malicious Use Evaluation | Capabilities Correlation | |
|---|---|---|
| | Base | Chat |
| **HarmBench DR** | | |
| Biochemical | -0.54 | -0.04 |
| Cybercrime | -0.50 | -0.07 |
| Harassment | -0.45 | -0.16 |
| Harmful | -0.42 | 0.24 |
| Illegal | -0.41 | 0.09 |
| Misinfo | -0.44 | -0.37 |
| **WMDP** | | |
| WMDP Bio | -0.91 | -0.87 |
| WMDP Chem | -0.88 | -0.86 |
| WMDP Cyber | -0.86 | -0.87 |
| **CybersecEval2** | | |
| Autocomplete | -0.74 | -0.77 |
| Exploit | -0.31 | -0.49 |
| Instruct | -0.43 | -0.90 |
| MITRE | -0.25 | 0.55 |
| Prompt Injection | -0.02 | -0.17 |

# 5  Discussion

As our experiments show, safety and capabilities metrics can be intertwined, with some safety metrics improving naturally as a consequence of general capabilities advancements. We first summarize high-level patterns in the results, then discuss several implications of our findings.

**Patterns across safety areas.** Our capabilities correlation analysis reveals varied dependencies on model scaling across safety areas. Adversarial robustness exhibits mixed results, with some perturbation robustness tasks improving with general capabilities while jailbreak robustness remains uncorrelated with general capabilities. Bias measures show some improvement with capabilities, but many biases are unaffected, suggesting that scaling alone may not mitigate all bias issues. Machine ethics generally correlates positively with capabilities, though further refinement is needed for balanced ethical understanding. In malicious use and rogue AI risk domains, we observe negative correlations, indicating a tendency for these risks to worsen with scale unless mitigated by specific interventions. These insights guide safety research toward areas where scaling alone is insufficient for safety progress.

**The importance of low capabilities correlation.** While many datasets measure interesting aspects of safety, these aspects are often not unique and instead are highly correlated with general capabilities. We argue researcher time for developing methods should be allocated toward solving benchmarks that won't be solved with scale and general capabilities advancements. Thus, capabilities correlation can be used as a metric for identifying which problems to spend research effort making progress on. If capabilities correlation for a benchmark is low, this means it will likely require additional algorithmic effort to make progress on.

**Measuring properties that improve with scale is still valuable.** Even if an evaluation is expected to be solved with scale, it is still useful to measure it. For instance, knowing the dangerous capabilities that emerge with scale is crucial. Reporting how evaluations scale with model size can help predict future risks, particularly with dangerous capabilities. This information is highly relevant, as it can indicate which problems may worsen with scale. The goal is not to measure the usefulness of a safety dataset, but to understand how to allocate research efforts efficiently. Evaluations showing strong correlation or anti-correlation with capabilities are valuable for tracking the evolution of dangerous capabilities and ensuring precise measurement of safety metrics, even if they are eventually solved with scale. Furthermore, there are some cases in our evaluations where tasks with high capabilities correlations will be improved but not completely "solved" as general capabilities improve. However, in some cases, safety-related benchmarks may act a jangle for capabilities; jangle fallacy is the erroneous belief that two constructs are different because they have the different names, when in practice they measure the same latent factor.

**Improving capabilities correlation as a goal for safety research.** If a meaningful safety metric is strongly correlated with general capabilities, this is a good outcome, because it means the problem may be largely addressed by scaling even if present-day models struggle. As a corollary, safety research should seek to develop new methods and model classes that cause safety metrics to correlate more strongly with capabilities. However, this should not be taken too far. Past a certain threshold, further efforts to align safety metrics with general capabilities are unnecessary. Once this is achieved, research efforts can be re-allocated elsewhere. Moreover, our results show that in some cases a high capabilities correlation may not be sufficient to ensure that a safety property is fully achieved. In these cases, continued effort on developing differential safety improvements is warranted.

**New safety evaluations should report their correlation with capabilities.** This practice can ensure that evaluations initially measure a meaningful safety property that requires research effort to improve, rather than simply increased scale. This is notably done in some past papers, such as RuLES [54] and EQ Bench [59]. For example, it can be useful to know if a malicious use benchmark worsens with scale. A low correlation does not necessarily imply that the safety metric is irrelevant; it may indicate flaws in the dataset, such as insufficient data to detect small changes in progress, or that the dataset measures a different aspect entirely.

**Broad application of correlation analysis.** This analysis can be applied in a broad range of scenarios when determining whether an evaluation measures a meaningfully different property. In a broad range of scenarios, it may be the case that a confounding variable that better explains performance, rather than what a benchmark claims it measures. Future investigations can also investigate correlations of safety with various types of capabilities; while previous research already shows that performance

on different categories of capabilities (such as reasoning, knowledge, coding, or mathematics) seem to be tied to scale, other papers have found that reasoning and knowledge can represent different components of a PCA analysis [21].

**Recommendations.** Our recommendations are as follows:

1. For benchmark selection (and broadly, research problem selection), researchers should allocate more time toward creating and climbing safety benchmarks with low correlation and thus represent problems that will not be solved with general capabilities advancements using current methods.

2. For technique development, a fruitful research direction is to develop new safety methods that increase the capabilities correlation, ensuring that the safety benchmarks will improve in the future as general capabilities improve.

## 6   Conclusion

We have shown that a wide variety of safety benchmarks are tightly correlated with general model capabilities, calling their importance into question. By considering the Bitter Lesson and the continued scaling of deep learning, we argued that research in AI safety should anticipate the possibility of safety metrics being correlated with general capabilities, such that they are naturally solved with scale. To quantify this, we developed a methodology to measure the correlation of safety metrics with general capabilities via spectral analysis of accuracy on capabilities datasets. In experiments, we measured the capabilities correlation of a wide variety of safety benchmarks and datasets, finding that many prior datasets are strongly correlated with general capabilities. We make two specific recommendations: future safety benchmarks should aim for low correlation with general capabilities, while future safety methods should aim to increase correlation between relevant safety metrics and general capabilities.

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

# A  Appendix

## A.1  List of Models

### A.1.1  List of Language Models

The following list are all of the base models we used for our evaluations.

1. gemma-2B [84]
2. gemma-7B [84]
3. Llama-2-7B [87]
4. Llama-2-13B [87]
5. Llama-2-70B [87]
6. Llama-3-8B [85]
7. Llama-3-70B [85]
8. Mistral-7B-v0.1 [38]
9. Mixtral-8x7B-v0.1 [39]
10. falcon-7B [3]
11. falcon-40B [3]
12. falcon-180B [3]
13. Yi-6B [2]
14. Yi-9B [2]
15. Yi-34B [2]
16. Qwen1.5-0.5B [7]
17. Qwen1.5-1.8B [7]
18. Qwen1.5-4B [7]
19. Qwen1.5-7B [7]
20. Qwen1.5-14B [7]
21. Qwen1.5-32B [7]
22. Qwen1.5-72B [7]
23. deepseek-llm-7B-base [18]
24. deepseek-llm-67B-base [18]

The following list are all of the chat or instruct models we used for our evaluations.

1. gemma-1.1-2B-it [84]
2. gemma-1.1-7B-it [84]
3. Llama-2-7B-Chat [87]
4. Llama-2-13B-Chat [87]
5. Llama-2-70B-Chat [87]
6. Llama-3-8B-Instruct [85]
7. Llama-3-70B-Instruct [85]
8. Mistral-7B-Instruct-v0.2 [38]
9. Mixtral-8x7B-Instruct-v0.1 [39]
10. falcon-7B-Instruct [3]
11. falcon-40B-Instruct [3]
12. Yi-6B-Chat [2]
13. Yi-34B-Chat [2]
14. Qwen1.5-1.8B-Chat [7]
15. Qwen1.5-4B-Chat [7]
16. Qwen1.5-7B-Chat [7]
17. Qwen1.5-14B-Chat [7]
18. Qwen1.5-32B-Chat [7]
19. Qwen1.5-72B-Chat [7]
20. Qwen1.5-110B-Chat [7]
21. deepseek-llm-7B-Chat [18]
22. deepseek-llm-67B-Chat [18]

Note that all of the model names above are as you would find them on HuggingFace.

### A.1.2  List of Vision Models

The following is the list of vision models used in our evaluations in appendix A.4.

1. ResNet50 [78]
2. ResNet50 + $L_2$ 0.01 [78]
3. ResNet50 + $L_2$ 0.03 [78]
4. ResNet50 + $L_2$ 0.05 [78]
5. ResNet50 + $L_2$ 0.1 [78]
6. ResNet50 + $L_2$ 0.25 [78]
7. ResNet50 + $L_2$ 0.5 [78]
8. ResNet50 + $L_2$ 1 [78]
9. ResNet50 + $L_2$ 3 [78]
10. ResNet50 + $L_2$ 5 [78]
11. ResNet50 + $L_\infty$ 0.5/255 [78]
12. ResNet50 + $L_\infty$ 1.0/255 [78]
13. ResNet50 + $L_\infty$ 2.0/255 [78]
14. ResNet50 + $L_\infty$ 4.0/255 [78]

15. ResNet50 + $L_\infty$ 8.0/255 [78]

16. WideResNet-50-2 [78]

17. WideResNet-50-2 + $L_2$ 0.01 [78]

18. WideResNet-50-2 + $L_2$ 0.03 [78]

19. WideResNet-50-2 + $L_2$ 0.05 [78]

20. WideResNet-50-2 + $L_2$ 0.1 [78]

21. WideResNet-50-2 + $L_2$ 0.25 [78]

22. WideResNet-50-2 + $L_2$ 0.5 [78]

23. WideResNet-50-2 + $L_2$ 1 [78]

24. WideResNet-50-2 + $L_2$ 3 [78]

25. WideResNet-50-2 + $L_2$ 5 [78]

26. WideResNet-50-2 + $L_\infty$ 0.5/255 [78]

27. WideResNet-50-2 + $L_\infty$ 1.0/255 [78]

28. WideResNet-50-2 + $L_\infty$ 2.0/255 [78]

29. WideResNet-50-2 + $L_\infty$ 4.0/255 [78]

30. WideResNet-50-2 + $L_\infty$ 8.0/255 [78]

31. WideResNet-50-4 [78]

32. WideResNet-50-4 + $L_2$ 0.01 [78]

33. WideResNet-50-4 + $L_2$ 0.03 [78]

34. WideResNet-50-4 + $L_2$ 0.05 [78]

35. WideResNet-50-4 + $L_2$ 0.1 [78]

36. WideResNet-50-4 + $L_2$ 0.25 [78]

37. WideResNet-50-4 + $L_2$ 0.5 [78]

38. WideResNet-50-4 + $L_2$ 1 [78]

39. WideResNet-50-4 + $L_2$ 3 [78]

40. WideResNet-50-4 + $L_2$ 5 [78]

41. ResNet18 [78]

42. ResNet18 + $L_2$ 0.01 [78]

43. ResNet18 + $L_2$ 0.03 [78]

44. ResNet18 + $L_2$ 0.05 [78]

45. ResNet18 + $L_2$ 0.1 [78]

46. ResNet18 + $L_2$ 0.25 [78]

47. ResNet18 + $L_2$ 0.5 [78]

48. ResNet18 + $L_2$ 1 [78]

49. ResNet18 + $L_2$ 3 [78]

50. ResNet18 + $L_2$ 5 [78]

51. ResNet18 + $L_\infty$ 0.5/255 [78]

52. ResNet18 + $L_\infty$ 1.0/255 [78]

53. ResNet18 + $L_\infty$ 2.0/255 [78]

54. ResNet18 + $L_\infty$ 4.0/255 [78]

55. ResNet18 + $L_\infty$ 8.0/255 [78]

56. ResNeXt-50 32x4d [78]

57. ResNeXt-50 32x4d + $L_2$ 3 [78]

58. DenseNet [78]

59. DenseNet + $L_2$ 3 [78]

60. ShuffleNet [78]

61. ShuffleNet + $L_2$ 3 [78]

62. VGG16BN [78]

63. VGG16BN + $L_2$ 3 [78]

64. MnasNet [78]

65. MnasNet + $L_2$ 3 [78]

66. MobileNet [78]

67. MobileNet + $L_2$ 3 [78]

68. DINOv2 ViT-large Patch14 [57]

69. ConvNeXt-V2-large ImageNet1K+22K [98]

70. DINOv2 ViT-base Patch14 [57]

71. ViT-base Patch16 + $L_\infty$ 4/255 [80]

72. Swin-large ImageNet1K [47]

73. ConvNeXt-V2-huge ImageNet1K [98]

74. Swin-base ImageNet1K [47]

75. ConvNeXt-V2-base ImageNet1K+22K [98]

76. ConvNeXt-xlarge ImageNet1K+22K [48]

77. ConvNeXt-V2-large ImageNet1K [98]

78. MAE ViT-large Patch16 [26]

79. ConvNeXt-base + $L_\infty$ 4/255 [48]

80. ViT-small Patch16 + $L_\infty$ 4/255 [80]

81. ConvNeXt-large ImageNet1K+22K [48]

82. Swin-base ImageNet1K + $L_\infty$ 4/255 [47]

83. ConvNeXt-small + $L_\infty$ 4/255 [48]

84. ViT-base Patch8 ImageNet1K+22K [80]

85. Swin-small ImageNet1K + $L_\infty$ 4/255 [47]

86. ConvNeXt-base ImageNet1K+22K [48]

87. ConvNeXt-V2-base ImageNet1K [98]

88. ViT-large Patch16 ImageNet1K+22K [80]

89. ConvNeXt-large ImageNet1K [48]

90. Swin-small ImageNet1K [47]

91. ConvNeXt-V2-tiny ImageNet1K+22K [98]

92. ConvNeXt-small ImageNet1K+22K [48]

93. ConvNeXt-base ImageNet1K [48]

94. CLIP (ViT-L/14) [69]

95. ConvNeXt-V2-tiny ImageNet1K [98]

96. MAE ViT-base Patch16 [26]

97. ConvNeXt-small ImageNet1K [48]

98. ResNet50 + $L_\infty$ 8/255 [78]

99. ConvNeXt-V2-nano ImageNet1K [98]

100. ConvNeXt-V2-nano ImageNet1K+22K [98]

101. Reversible-ViT-base multiscale [49]

102. ViT-base Patch16 ImageNet1K+22K [80]

103. ConvNeXt-tiny ImageNet1K+22K [48]

104. ConvNeXt-tiny ImageNet1K [48]

105. Swin-tiny ImageNet1K [47]

106. ResNet50 + PixMix [78]

107. ResNet50 + Moex [78]

108. Reversible-ViT-base [49]

109. ResNet50 + CutMix [78]

110. Reversible-ViT-small [49]

111. ConvNeXt-V2-pico ImageNet1K [98]

112. ConvNeXt-V2-atto ImageNet1K [98]

113. ResNet50 + $L_\infty$ 4/255 [78]

114. ResNet50 + DeepAug+AugMix [78]

115. ResNet50 + Mixup [78]

116. ResNet50 + Deepaugment [78]

117. ResNet50 + $L_\infty$ 2/255 [78]

118. ViT-base Patch16 ImageNet1K [80]

119. ConvNeXt-V2-femto ImageNet1K [98]

120. ResNet50 + $L_\infty$ 1/255 [78]

121. ViT-small Patch16 ImageNet1K [80]

122. ViT-small Patch16 ImageNet1K+22K [80]

123. ViT-base Patch32 ImageNet1K+22K [80]

124. ViT-base Patch32 ImageNet1K [80]

125. ResNet50 + AugMix [78]

126. ResNet50 + Stylised ImageNet [78]

127. ResNet50 + ANT [78]

128. ResNet50 + RandAug [78]

129. ViT-small Patch32 ImageNet1K+22K [80]

130. ViT-tiny Patch16 ImageNet1K+22K [80]

## A.2 Closed Source Model Evaluations

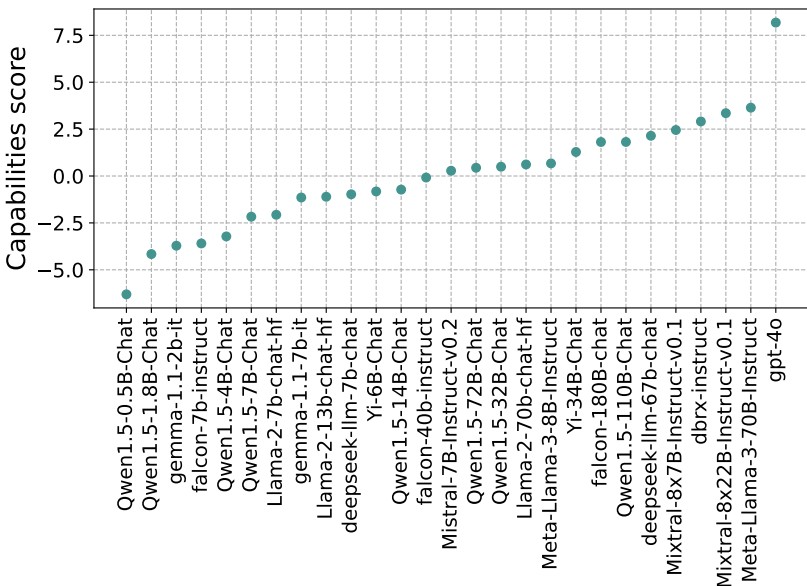

Figure 8: Capabilities scores for all the open source instruction-tuned and chat models we evaluated, plus GPT-4o.

In this section we compare the capabilities score of GPT-4o, today one of the most capable closed source models, with other open source LLMs we evaluated. We compute the capabilities scores by evaluating the models on the full capabilities set. The correlation analysis is then performed on this matrix of scores. Table 3 contains the scores of GPT-4o on the capabilities tasks, while in Figure 8 we can observe that the capabilities scores roughly grow with the training FLOP of a language model, with a clear gap between GPT-4o and the current open source models.

Table 3: Evaluation of GPT-4o on different capabilities tasks. Columns report the relevant metric (in percentages) on the corresponding dataset.

|                     | GPT-4o |
|---------------------|--------|
| MMLU (full)         | 84     |
| HellaSwag           | 92     |
| ARC-Challenge       | 94     |
| LogiQA              | 58     |
| PIQA                | 96     |
| WinoGrande          | 84     |
| SuperGLUE (copa)    | 100    |
| MedQA (4 options)   | 87     |
| MATH                | 83     |
| GMS8K               | 69     |
| BBH                 | 83     |

## A.3 Calibration

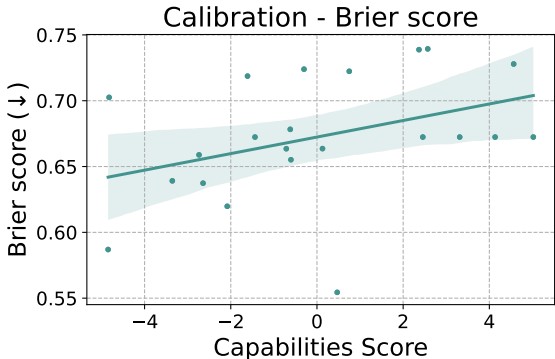

Figure 9: We observe a slightly positive correlation between the Brier score and that of capabilities score across chat models indicating that models tend to be less calibrated as capabilities increases.

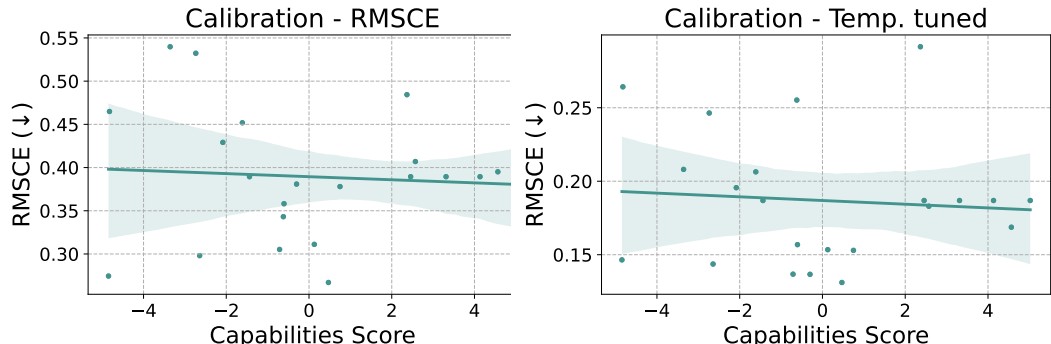

Figure 10: We observe that there exists no correlation between chat models' capabilities and calibration error. Temperature tuning beneficially acts to improve RMSCE regardless of model capability.

Calibration in machine learning models refers to how well the predicted probabilities of a model align with the actual probabilities of the observed outcomes. A well-calibrated model will produce predictions where the confidence levels correspond accurately to the likelihood of correctness. There exist a variety of methods such as temperature tuning, platt scaling, and isotonic regression to better calibrate models.

Overall, we observe that calibration as a research direction is generally uncorrelated with model capabilities. Figure 10 illustrates this lack of correlation, showing that advancements in model capabilities do not necessarily lead to improvements in calibration. This indicates that even as models become more sophisticated and accurate, their ability to produce well-calibrated probabilities does not automatically improve. We can also observe that techniques such as temperature tuning help calibrate all models roughly equally by dropping the mean error rate from 39% to 18%.

In some cases, particularly when considering the Brier score, calibration can even worsen as model capabilities increase. Figure 9 presents a slightly positive trend line, suggesting that as models become more capable, their calibration might deteriorate. This phenomenon can be attributed to overconfidence in predictions, where highly capable models assign higher probabilities to their predictions, which do not always correspond to the actual frequencies of the outcomes. As a result, despite improved accuracy, the reliability of the predicted probabilities may suffer, highlighting the need for continued research and development in the calibration of advanced models.

## A.4 Adversarial Robustness in Vision Models

Adversarial robustness remains a significant challenge in the realm of computer vision. Models trained on standard datasets often exhibit vulnerabilities to adversarial inputs—carefully crafted perturbations that, while imperceptible to humans, can lead to substantial misclassification errors and degradation in model performance. These adversarial examples exploit the weaknesses in the models' decision boundaries, revealing a gap in their ability to generalize beyond the training distribution. Robustness in this context refers to a model's ability to maintain high accuracy despite the presence of such adversarial perturbations.

Figure 11 investigates the correlation between an image classification model's standard accuracy and its capability, as measured by the model's robustness and capability score computed from ImageNet-O [31] and ImageNet-A [31], which are datasets designed to evaluate how well models generalize to out-of-distribution samples and handle naturally occurring adversarial examples, respectively. For robust models, we observe a strong correlation between the standard accuracy and robust accuracy. Furthermore, for both standard and robust models, there is a strong correlation between the standard accuracy and the capability score computed from ImageNet-A and ImageNet-O. These findings are consistent with the results obtained from language models, suggesting that the relationship between standard accuracy and model capability extends across different domains. On the other hand, enhancements in model capabilities do not inherently address the problem of adversarial robustness. This is indicated from the fact that all standard models perform poorly against these ad-

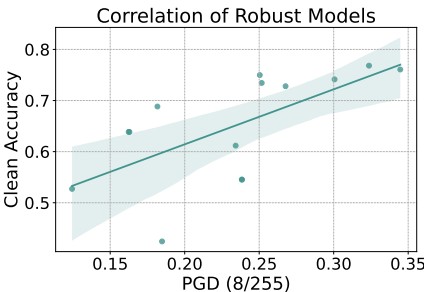

Figure 11: Capabilities correlations on standard/adversarially robust vision models (see Appendix A.1 for the full list of vision models; the adversarially trained models only consists of models trained with $L_\infty$ 4/255 or $L_\infty$ 8/255). The capability scores are computed from two benchmarks: ImageNet-A [31] and ImageNet-O [31].

versaries. In addition, in contrast to language models adversaries, the attack space for vision models might be larger due to the continuous nature of the input, making the problem more challenging.

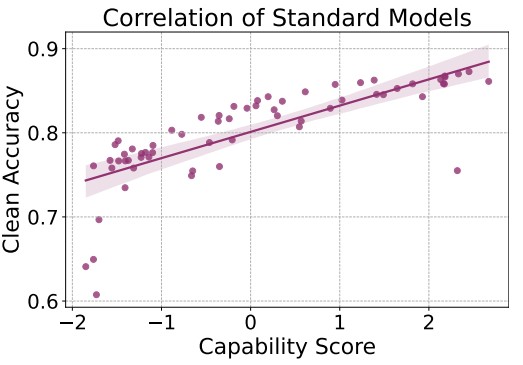
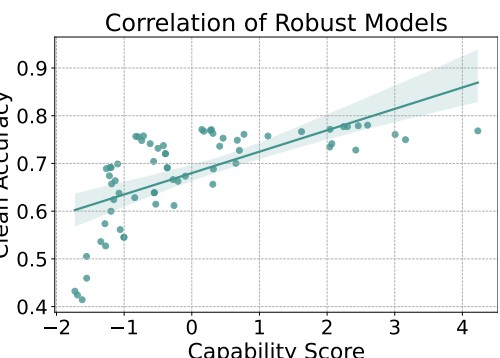

Figure 12: Correlation between clean accuracy and capabilities for both standard and adversarially robust models. In both cases, more capable models also show better accuracy performance.

## A.5 Hallucinations

Hallucinations refer to instances where language models generate content that is plausible-sounding but factually incorrect or nonsensical. It is mildly well-known that as language models scale up, they tend to hallucinate less [36, 45]. We further verify this hypothesis by measuring the capabilities coefficient on a generative benchmark (TruthfulQA Generation [45]) and a discriminative benchmark (HaluEval [43]). We report the capabilities correlation of these hallucination benchmarks in Table 4.

**Hallucinations decrease as capabilities increase.** Our analysis shows that as models' general capabilities improve, the tendency for hallucinations decreases. This correlation is observed in both generative and classification tasks across both base and chat models.

**Chat models improve overall truthfulness but with lower capabilities correlation than base models.** Our findings indicate that while instruction-tuned or chat models generally produce more truthful responses compared to their base models, the correlation between capabilities and truthfulness is relatively lower for chat models. As depicted in Figure 13, although chat models achieve higher accuracy for TruthfulQA generation, they exhibit a weaker correlation between their capabilities and the generation of truthful responses compared to base models.

Table 4: Capabilities correlations for Hallucinations and Truthfulness datasets. In general, instruction-tuned and chat models show lower correlation with capabilities compared to base models.

| Hallucinations | Capabilities Correlation | |
| --- | --- | --- |
| | Base | Chat |
| **HaluEval** [43] | | |
| HaluEval All | 0.70 | 0.53 |
| HaluEval Summarization | 0.41 | 0.25 |
| HaluEval Dialogue | 0.74 | 0.79 |
| HaluEval QA | 0.38 | 0.20 |
| **TruthfulQA** [45] | | |
| TruthfulQA Generation | 0.73 | 0.55 |

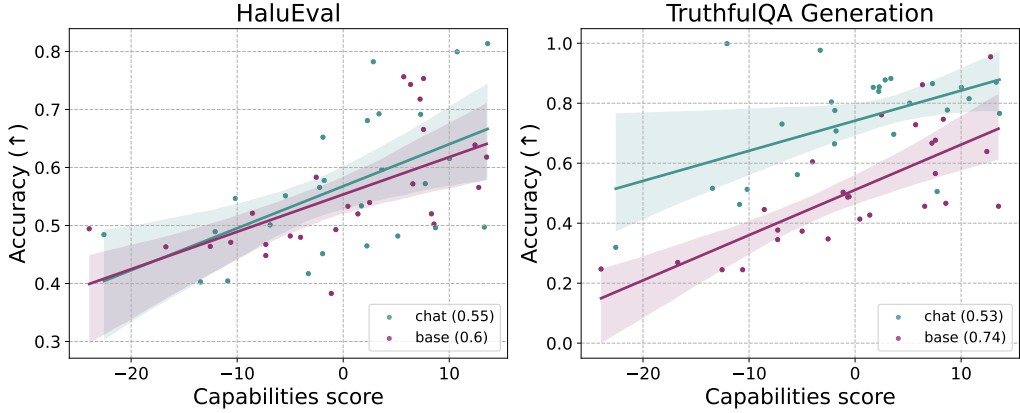

Figure 13: The capabilities score correlates with hallucination and truthfulness benchmarks for both base and chat models.

## A.6 Sensitivity analysis

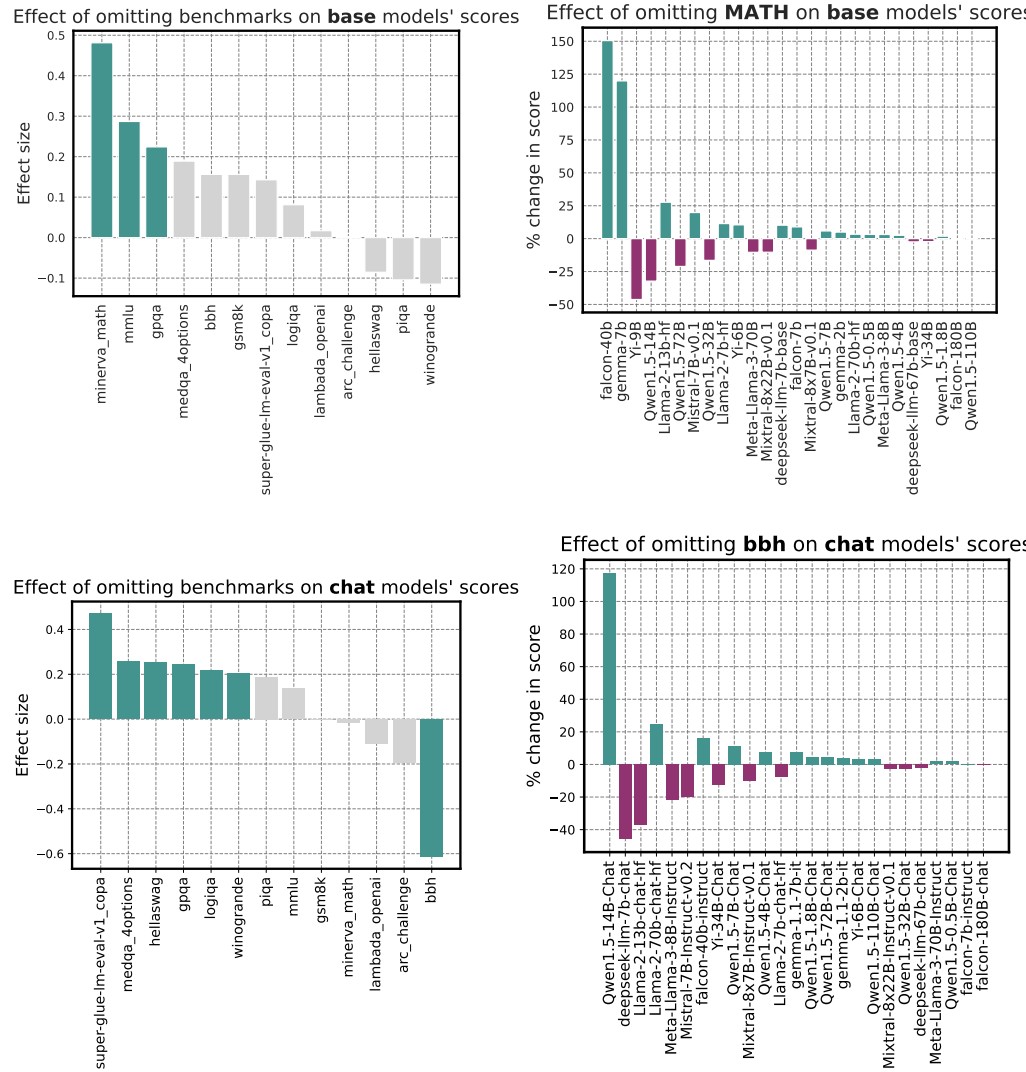

Figure 14: Leave One Out sensitivity analysis summary. The top row corresponds to base models and the bottom row to instruction tuned models. The left column figures show the effect size of removing a benchmark on model capabilities as measured by Cohen's d value on top five models in terms of absolute value of relative change in capabilities score. The gray benchmarks correspond to negligible effect. Green is used signify small and medium effects. The right column shows the impact (measured by signed percentage change) of removing the benchmark with highest effect on individual model capabilities scores.

Figure 14 shows a summary of our Leave One Out
analysis. We remove one benchmark at a time and repeat spectral analysis of model capabilities scores using only the remaining benchmarks. We then compute impacts of this omission on each model as a signed percentage change in capabilities scores. We also look at the shift in distribution of model capabilities after removing each benchmark and summarize the it by computing Cohen's $d$ – a statistical measure used to quantify the effect size or the magnitude of difference between two groups' means in relation to their combined standard deviation. Cohen's $d$ provides a standardized way to understand the size of an effect (e.g. the impact of an intervention) beyond the mere statistical significance of the difference, offering insights into its practical importance. Cohen's $d$ is calculated as

the difference between two means divided by the pooled standard deviation of the data. Positive value indicates that the scores increase in the absence of the chosen task, while negative value indicates that the scores decrease.

In our Leave One Out sensitivity analysis we are interested in whether the intervention of removing a particular task has a significant effect on model performance on capabilities in general. We use the following generally accepted heuristic: benchmarks with effect size score (Cohen's d) of less than 0.2 are considered to have a negligible effect, those with scores between 0.2 and 0.5 have medium effect, and those with scores above 0.5 have large effect.

Among base models no task had medium or higher effect. Three tasks: Minerva Math, MMLU, GPQA had small effect (sorted the order of decreasing score). We also show the impact of removing the top task (Minerva Math) on each base model as a signed percentage change in capabilities scores (see the lower left subplot in figure 14). Two models stand out in this comparison as being disproportionately affected by this task: falcon-40b and gemma-7b.

Chat models seemed more sensitive to removing tasks. Six tasks had small effect: super-glue, medqa, hellaswag, GPQA, logiqa, winogrande. There was also one task with a medium size (negative) effect: BBH. Lower right subplot in figure 14 shows the impact of removing the top task (BBH) on each base model as a signed percentage change in capabilities scores.

### A.7 Compute Environment

The experiments were conducted using the BM.GPU4.8 bare metal instance on Oracle Cloud. This setup includes 8 NVIDIA A100 Tensor Core GPUs, each with 80 GB of memory, interconnected via NVIDIA NVLink. The instances use an AMD Rome processor with 64 physical cores operating at 2.9 GHz. It also features 2,048 GB of RAM, 24 TB of NVMe storage, and offers a network throughput of 1.6 Tbps.

### A.8 Limitations

While our work provides valuable insights into the correlation between scaling laws and safety-specific properties, it is not without limitations. Our work does not discuss all issues that safety evaluations face. For example, data diversity and biases can affect the generalizability of results, which are separate concerns from correlation of a safety metric with general capabilities. Another limitation is that our analysis does not encompass all possible safety benchmarks or capabilities metrics, potentially overlooking certain nuances specific to individual datasets or models. Additionally, while we emphasize the importance of distinguishing safety improvements from general capabilities enhancements, this should not discourage research in areas where these aspects are correlated. Understanding and documenting these correlations can still yield significant insights and advance the field by making these connections more explicit such as in WMDP [44]. Thus, while our findings highlight the need for targeted safety evaluations, they also underscore the broader relevance of exploring how safety and capability factors interplay in AI development.

### A.9 Broader Impacts

The broader impacts of this work are significant, as it addresses critical gaps in understanding the relationship between model scaling and safety. By elucidating how scaling laws correlate with safety-specific properties, this research provides a framework for developing AI systems that are not only more capable but also inherently safer. This dual focus on capability and safety is essential for the responsible deployment of AI technologies in various high-stakes environments, including healthcare, autonomous driving, and finance. Enhancing the robustness and reliability of AI systems through targeted safety improvements can help mitigate risks associated with AI deployment, thereby protecting users and stakeholders from potential harms.

Furthermore, this work has implications for the future design and evaluation of AI benchmarks. By proposing a more nuanced approach to weighting benchmark components based on their variance in model performance, we encourage a shift towards more comprehensive and meaningful evaluations of AI systems. This could lead to the development of new benchmarks that better capture the multifaceted nature of AI capabilities and safety, driving innovation in both areas. Additionally, our findings could influence regulatory and policy frameworks, providing evidence-based guidelines for

assessing and ensuring AI safety. In the long term, this research contributes to the broader goal of creating AI systems that are not only powerful but also aligned with human values and societal needs.

## A.10    Size Is Not Always Correlated with Capabilities

To further investigate the relationship between model size and capabilities, we conducted experiments with the Llama 3.1 model series. Our experiments revealed instances where size did not correlate positively with performance on certain benchmarks.

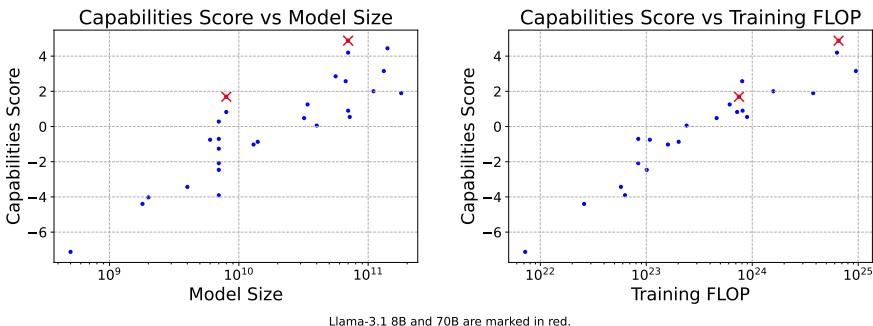

Figure 15: Performance of Llama 3.1 Models Across Capabilities Benchmarks.

We evaluated multiple versions of the Llama 3.1 models, varying in parameter count and training configurations. The models were assessed on a range of capabilities and safety benchmarks to determine how changes in size impacted their performance.

Our experiments demonstrated that increasing model size does not guarantee improved performance across all tasks. In some cases, smaller models outperformed their larger counterparts on specific safety benchmarks. This suggests that factors other than size, such as training data diversity, optimization techniques, and architectural choices, play crucial roles in determining a model's capabilities. We also found that training FLOP still remained highly correlated with capabilities.

## A.11    Capabilities Correlations for Knowledge, Reasoning, and Math

Building upon our primary analysis, we explored how the correlations between safety benchmarks and model capabilities change when focusing on specific types of capabilities: mathematical reasoning, general knowledge, and common sense.

We recalculated the capabilities component scores using subsets of benchmarks corresponding to each capability type:

- **Math Capabilities Component**: MATH [30], GSM8K [17]
- **Knowledge Capabilities Component**: MMLU [29], MedQA [40], ARC Challenge [16]
- **Common-Sense Capabilities Component**: LogiQA [46], PIQA [13], HellaSwag [101], Winogrande [77], COPA [73], LAMBADA [62], Big Bench Hard [9]

Using these components, we calculated the correlation coefficients between the safety benchmarks and each capability type.

The results, as shown in Table 5, indicate that while general patterns persist, the correlations between safety benchmarks and capabilities can vary significantly depending on the type of capability considered. For example:

- **Knowledge Component**: Exhibited the highest positive correlations with safety benchmarks like AdvGLUE, AdvDemonstration, ETHICS, and TruthfulQA MC1.
- **Math Component**: Showed moderate correlations, suggesting that mathematical reasoning contributes differently to safety benchmark performance.

Table 5: Correlation Coefficients Between Safety Benchmarks and Capabilities Components

| Evaluation | Math | Knowledge | Common Sense |
|---|---|---|---|
| AdvGLUE | 37.0 | 70.8 | 59.9 |
| AdvGLUE++ | 35.5 | 65.2 | 37.8 |
| AdvDemonstration | 47.8 | 80.0 | 54.8 |
| HarmBench Human | -22.4 | -37.8 | -29.6 |
| HarmBench TAP-T | -31.5 | -37.9 | -41.5 |
| HarmBench GCG-T | -18.8 | -33.9 | -23.8 |
| CrowS-Pairs English | 31.1 | -2.3 | 36.3 |
| Discrim-Eval | 43.2 | 45.6 | 26.0 |
| ETHICS | 61.8 | 83.1 | 85.0 |
| Sycophancy | -66.8 | -68.6 | -57.5 |
| TruthfulQA MC1 | 61.1 | 86.0 | 77.0 |
| WMDP Bio | -68.9 | -95.0 | -78.7 |
| WMDP Chem | -65.4 | -93.3 | -71.0 |
| WMDP Cyber | -64.8 | -96.1 | -77.1 |
| RuLES | 36.6 | 40.7 | 39.7 |

- **Common Sense Component**: Correlations were generally positive but varied more than the knowledge component.

- **Negative Correlations**: Benchmarks like HarmBench and WMDP showed strong negative correlations across all capability types, indicating that increased capabilities in these areas might exacerbate certain safety risks.

