# OpenReview forum: "Safetywashing: Do AI Safety Benchmarks Actually Measure Safety Progress?"
_NeurIPS.cc/2024/Datasets_and_Benchmarks_Track — NeurIPS 2024 Track Datasets and Benchmarks Poster_

### Official Review · Reviewer_DMFp · 2024-06-24
**Beneficial contribution, could improve clarity.**

**Rating:** 7
**Confidence:** 5

**Review:**

Practically, the contributions of this work can be split into two separate "principal" components (pun intended):
- A holistic and thoughtful suggestion for best process and directions for progress to the AI safety community
- A method for quantifying the degree to which a given benchmark scales with overall model capabilities

My broadest concern is that **the paper should do a better job of making the distinction between these two contributions clear**. Since it is not clearly drawn, it makes interpretation of the contributions more difficult. Specifically, Section 5, Discussion, consists entirely of broad recommendations to the safety community, and is completely unrelated to the results outlined in Section 4.

This is unfortunate, as due to the size and organization of Section 4, it is difficult to draw a clear conclusion and takeaways from the results given in Section 4 itself. So, the paper would benefit greatly from a discussion of these results, such as an overarching table or list detailing all the categories of safety benchmarks and their relation to overall capabilities.

Below I detail some additional strengths and weaknesses more specifically.

**Strengths:**

- Fundamentally, and most importantly, **I do believe that the broad message and suggestions within the paper are a valuable contribution to the research community**.
- The variety of safety benchmarks evaluated and presented of results is thorough.

**Additional Feedback:**

Happy to increase the score once my concerns (especially over writing) are addressed.

[Update]: Change score from 6 to 7.

**Clarity:**

Overall, many parts of the paper, especially those concerning the greater messaged, are clear and well written. However, as mentioned, the greatest issue with clarity is in the relationship between the proposed method and the greater message. And, in particular, the paper could benefit from a discussion that actually incorporates the results.

Specifically: Given the overall message that the authors want to make to the community **how do their presented results actually inform/augment/coincide with this message**?

The results themselves could also be presented clearer. The results section contains a large number of figures that each a small part of the study, but it is hard to piece together the bigger picture of the results section. There are also a very large amount of raw numbers that as hard to conceptualize of.

**Correctness:**

There is not reason to believe that the authors were disingenuous in their results.

**Documentation:**

N/A

**Ethics:**

No.

**Limitations:**

The authors address limitations in the appendix - a small note, the authors suggest in the checklist that they also discuss limitations in the discussion section, but they don't seem to do so (not a big issue).

**Opportunities For Improvement:**

The capabilities component is meant to be a "distillation" of many different popular LLM benchmarks such as MMLU and MedQA. But what does this component actually measure? It is unclear to me how overall capabilities can be distilled into a single measurement, as models might exhibit different specific capabilities. And, the included benchmarks themselves measure a variety of different capabilities, such as knowledge, retrieval ability, reasoning, etc. So, **what does this component actually measure?**

This kind of fine-grained study of capabilities might actually be important for the study conducted in this paper, as different safety benchmarks might actually be correlated with *specific* capabilities. For example, a safety benchmark measuring dangerous information might be correlated specifically with knowledge, but might be uncorrelated with reasoning ability.

The only analysis of the capabilities component I see in the paper is showing that is correlated to training flops, but almost any metric of any model capabilities would have this behavior. **I believe more justification and analysis of this specific instantiation of a capabilities component is needed.**

A second note - the authors suggest that it would be naive to just average benchmark results to create the capabilities component, but it could be useful to see what this would really do.

**Relation To Prior Work:**

The authors make clear the novelty of their work.

**Summary And Contributions:**

This paper largely consists of a logical suggestion to the AI safety community: to consider the correlation between safety benchmarks and overall model capabilities when evaluation and development of safety methods. This guidance is largely based on the intuition that since we expect models too continue to scale, it is not worthwhile to address safety issue that will be might be solved "by themselves" as models scale, but rather we should identify and resolve safety concerns that will continue to persist or even worse.

This paper additionally introduced a method for quantifying the correlation between a given (safety) benchmark and overall model capabilities based on "distilling" current LLM capability benchmark into a capabilities component (via PCA on a benchmarks x models matrix), then comparing the new benchmark's score's to this component.

---

> ### Author Rebuttal · Authors · 2024-08-15
>
> Thank you for your detailed analysis of our paper. We hope the following response addresses your concerns.
>
> **Making the distinction between main contributions**
>
> As you note, two major contributions in our paper are a set of recommendations for AI safety researchers and the method for capabilities correlation analysis that we propose. We agree that clarifying the distinction between these contributions would help readers. In the updated paper, we will highlight these two contributions. In the discussion, we will add a section summarizing major takeaways from the results (e.g. which benchmarks were correlated, etc.), and then transition to a separate subsection organizing our recommendations in one place. Thank you for your helpful suggestion.
>
> **Clarifying what the capabilities component measures**
>
> > It is unclear to me how overall capabilities can be distilled into a single measurement, as models might exhibit different specific capabilities
>
> We agree that different models may exhibit different capabilities. Our single measurement was inspired by our empirical findings across models that models which are stronger on one capability tend to be stronger on other capabilities as well, suggesting the existence of underlying factors that determine the overall performance of a model (e.g., “knowledge”, “reasoning”, “experience”). For example, we find that 71% of the variance in benchmark scores in chat models can be explained by PC1 in our analysis.
>
> Inspired by your comments, we plan to report the results of an additional analysis to understand whether the correlations significantly change when we focus on one specific capability (i.e. calculate correlations using only knowledge only, retrieval ability only, mathematical reasoning only benchmarks). This analysis would elucidate whether the first principal component is only capturing one of these capabilities, without adequately capturing others. We hope that this analysis addresses your concerns.
>
> Besides this analysis, what this factor qualitatively corresponds to is somewhat of an open question. As you note, we find a very high correlation between scale and the capabilities score of a model, but other factors such as data and algorithms can have an effect on the overall performance of LLMs. Assigning specific high-level terms to the capabilities component, such as “knowledge” or “reasoning” may be beyond the scope of our work, but we think it could be an exciting direction for future work. We will add a discussion of this to the updated paper. Thank you for your suggestion.
>
> **Other points**
>
> > the authors suggest that it would be naive to just average benchmark results to create the capabilities component, but it could be useful to see what this would really do.
>
> We agree that this would be interesting information for the reader to see. We will add an experiment to the updated paper comparing the average score to PC1. Thank you for your suggestion.
>
> If we have addressed your concerns, we kindly ask that you consider raising your score.

---

> > ### Author Response · Authors · 2024-08-23
> > **Friendly reminder**
> >
> > This is just a friendly reminder that the discussion period is ending soon. If you have any followup questions about the paper or our reply to your initial concerns, we would be glad to answer them.

---

> > ### Comment · Reviewer_DMFp · 2024-08-24
> > **Increased Score**
> >
> > Thank you to the authors for addressing my issues. I am still slightly concerned about the use of the capabilities component (and look forward to additional analysis of this in the final work). However, given the overall message and potential impact of this work I have increased my rating.

---

> > ### Author Rebuttal · Authors · 2024-09-01
> >
> > We have finished running experiments mentioned in our reply related to whether safety benchmark correlations change significantly when focusing on one specific capability (e.g. mathematical reasoning). In the table below, we measure whether safety benchmarks’ correlations change when using only math, knowledge, or common-sense benchmarks in the capabilities component calculation:
> >
> > | **Evaluation** | **Math** | **Knowledge** | **Common Sense** |
> > |-----------------|----------|---------------|------------------|
> > | AdvGLUE | 37.0 | 70.8 | 59.9 |
> > | AdvGLUE++ | 35.5 | 65.2 | 37.8 |
> > | AdvDemonstration | 47.8 | 80.0 | 54.8 |
> > | HarmBench Human | -22.4 | -37.8 | -29.6 |
> > | HarmBench TAP-T | -31.5 | -37.9 | -41.5 |
> > | HarmBench GCG-T | -18.8 | -33.9 | -23.8 |
> > | CrowS-Pairs English | 31.1 | -2.3 | 36.3 |
> > | Discrim-Eval | 43.2 | 45.6 | 26.0 |
> > | ETHICS | 61.8 | 83.1 | 85.0 |
> > | Sycophancy | -66.8 | -68.6 | -57.5 |
> > | TruthfulQA MC1 | 61.1 | 86.0 | 77.0 |
> > | WMDP Bio | -68.9 | -95.0 | -78.7 |
> > | WMDP Chem | -65.4 | -93.3 | -71.0 |
> > | WMDP Cyber | -64.8 | -96.1 | -77.1 |
> > | RuLES | 36.6 | 40.7 | 39.7 |
> > - Math capabilities component: MATH, GSM8K
> > - Knowledge capabilities component: MMLU, MedQA, ARC Challenge
> > - Common-sense capabilities component: LogiQA, PIQA, HellaSwag, Winogrande, COPA, Lambada, Big Bench Hard
> >
> > We find that while many general patterns remain similar, some correlations may change significantly depending on the instantiation of capabilities (e.g. the capabilities correlation of AdvGLUE, AdvGLUE++, AdvDemonstration seems to change significantly across different capabilities).
> >
> > Because of your thoughtful suggestion as well as our findings above, we will add discussion in our paper about different types of capabilities and how their correlations with safety benchmarks may differ.

---

### Official Review · Reviewer_6eSn · 2024-07-23
**Promising paper; I strongly recommend implementing improvements to framing during the discussion period**

**Rating:** 6
**Confidence:** 4

**Review:**

See below. The paper's framing exhibits significant weaknesses. This is a promising paper, and I strongly encourage the authors to implement improvements during the discussion period.

[NOTE: The score has been increased as a result of authors deeply engaging with feedback during rebuttal, and promising to make changes.]

**Strengths:**

The approach of using factor analysis (inspired by psychometrics research on the ‘g’ factor in human intelligence) to attempt to differentiate/correlate improvements that are task-specific from general improvements across all tasks is compelling.

The paper is quite well written.

**Additional Feedback:**

See above.

**Clarity:**

Yes, with the exception of the significant lack of clarity in the paper's framing.

**Correctness:**

The claims made in the submission appear correct, but evaluation of correctness is hampered by the weaknesses delineated above.

**Documentation:**

Yes.

**Ethics:**

No.

**Limitations:**

See above, the paper would strongly benefit from a more explicit articulation of the limitations of the study.

**Opportunities For Improvement:**

The paper’s quality is significantly weakened by its approach to defining and delineating the core constructs it is attempting to differentiate (“capabilities” vs “safety”).

The paper avoids providing a clear and specific definition of safety, relying instead on (a) a loose description “safety research tends to study empirical phenomena that are negative side effects of model deployment [78, 52, 54, 59, 47], are malicious use of models [75, 86, 35], or do not improve with scale [8, 39]”, and (b) on tangible safety benchmarks that the authors provide measurements for—such as ETHICS, AdvGLUE, HarmBench, CybersecEval2, etc. The level of clarity provided b) is enough to support the kind of correlational measurements the paper provides in section 4.

However, without much more explicit and rigorous effort in defining and measuring safety as a construct, framing the paper’s findings as contrasting safety and capabilities is misleading. For instance, why not instead hypothesize that the paper’s approach helps capture correlation or lack thereof between task-specific performance (equivalent to the generality of a specific benchmark) and general performance (correlated with performance across all tasks)? More generally, the loose definition of safety the authors provide is woefully inadequate for supporting a rigorous conceptualization of the construct(s) the paper purports to measure.

Taking a closer look at psychometrics research is strongly advisable. In particular, a well-established contrast in psychometrics research leveraging the ‘g’ factor (popularized by Carroll’s “Human cognitive abilities” 1993 book) is between variations in performance on 3 levels of generality: task-specific (roughly equivalent to the generality of a specific benchmark), domain-specific (could potentially be equivalent to ‘safety’ benchmarks, but significant challenges arise in providing well defined domain constructs), and general (correlated with performance on all tasks). (See Deary 2012 for an overview: https://www.annualreviews.org/doi/10.1146/annurev-psych-120710-100353) The authors may benefit from engaging more deeply parallel questions that arise in the context of their own project — in particular about the challenge of distinguishing benchmark-specific performance from performance within well-constructed domains.

I’d also strongly recommend engaging with work on construct reliability and validity issues in ML evaluation, such as Jacobs, Abigail Z., and Hanna Wallach. “Measurement and Fairness.” In Proceedings of the 2021 ACM Conference on Fairness, Accountability, and Transparency, 375–85. Virtual Event Canada: ACM, 2021. https://doi.org/10.1145/3442188.3445901.

For the purposes of the present review process, authors need to do a much more explicit job of conceptualizing the scope and limitations of their study’s approach to contrasting safety and capabilities.

**Relation To Prior Work:**

Yes.

**Summary And Contributions:**

The paper looks at correlations between general and “safety-specific” performance improvements. Against the background of work showing that performance improvements on seemingly specific tasks (such as scores on “fairness” or “adversarial robustness” benchmarks) often correlates with improvement on all tasks—such as the kind of improvements described by scaling laws”, the paper contends that the safety benchmarking community needs to pay stronger attention to “safety-specific” improvements that do not correlate with scale.

---

> ### Author Rebuttal · Authors · 2024-08-15
>
> Thank you for your detailed analysis of our paper. We hope the following response addresses your concerns.
>
> **We rely on the research community to identify safety properties**
>
> > without much more explicit and rigorous effort in defining and measuring safety as a construct, framing the paper’s findings as contrasting safety and capabilities is misleading
>
> We fully agree that a more rigorous definition of safety as a construct would clarify the distinction between safety and capabilities. Unfortunately, there is no broadly accepted definition of safety among ML researchers. Instead, ML researchers often simply call a research problem “safety research” if it intuitively seems like it could reduce some form of risk, and different researchers may disagree on this.
>
> Thus, rather than attempt to precisely define safety as a construct, we opt to measure various specific properties that have been described as safety properties by ML researchers (e.g., bias, adversarial robustness, ethical knowledge). Each of these properties is a distinct construct, and they are measured very differently, yet they are all commonly described as safety properties. Rather than attempt to rigorously identify what connects these different properties, we propose capabilities correlations as a tool for researchers to better understand how to allocate their time between problems that the research community already views as safety research.
>
> We plan to explicitly state all of the above points in our updated draft, since this was not clearly stated in our current draft. We thank you for bringing this counterpoint up.
>
> **Applicability of our analysis beyond safety benchmarks**
>
> > why not instead hypothesize that the paper’s approach helps capture correlation or lack thereof between task-specific performance (equivalent to the generality of a specific benchmark) and general performance (correlated with performance across all tasks)
>
> As you note, we can apply our approach to measure the correlation between any metric and general capabilities. We focus on safety metrics in our paper, because safety research is a particularly interesting setting for our analysis. However, there are many other use cases as well, some of which we highlight in the discussion section under the heading, “Broad application of correlation analysis”.
>
> **Engaging with work on construct reliability and validity issues in ML evaluation**
>
> > I’d also strongly recommend engaging with work on construct reliability and validity issues in ML evaluation, such as Jacobs, Abigail Z., and Hanna Wallach.
>
> Thank you for sharing the *Measurement and Fairness* paper – we agree with the main claim of the paper that the process of creating measurement can lead to mismatches between the theoretical understanding of a construct and how it is operationalized in practice. It does strongly relate to our work – which does essentially test various safety benchmarks for construct validity. We plan to add explicit commentary about construct validity and how a safety property’s operationalization can influence its correlation, citing this paper for reference.
>
> **Inclusion of different levels of generality in our analysis**
>
> > The authors may benefit from engaging more deeply parallel questions that arise in the context of their own project — in particular about the challenge of distinguishing benchmark-specific performance from performance within well-constructed domains.
>
> We agree that analyzing correlations across levels of generality would be informative. We currently structure our paper around different subdomains of safety research and individual benchmarks within those subdomains. Therefore, we are already able to study multiple levels of generality using our existing results. In the updated paper, we will add an analysis of which high-level areas are more or less correlated with general capabilities. Thank you for your suggestion.
>
> If we have addressed your concerns, we kindly ask that you consider raising your score.

---

> > ### Author Response · Authors · 2024-08-23
> > **Friendly reminder**
> >
> > This is just a friendly reminder that the discussion period is ending soon. If you have any followup questions about the paper or our reply to your initial concerns, we would be glad to answer them.

---

> > > ### Comment · Reviewer_6eSn · 2024-08-26
> > > **I am increasing my score as a result of the author's rebuttal. Thank you for engaging with the feedback.**
> > >
> > > I am increasing my score as a result of the author's rebuttal. Thank you for engaging with the feedback.

---

### Official Review · Reviewer_iAJj · 2024-07-24
**Useful insight with some disagreeable positioning**

**Rating:** 6
**Confidence:** 3
**Clarity:** Yes

**Review:**

Pros:

* Interesting work
* This is excellent: "Naive composite benchmarks usually weight their component tests equally, averaging test scores. A more principled approach can involve weighting component benchmarks according to the strength of their association with each other, with higher weight placed on benchmarks that account for greater variance in model performance across benchmarks."
* "New safety evaluations should report their correlation with capabilities." This would be interesting to see.

Cons:

* "...success in new safety methods should be measured not only by improvements in safety benchmark scores, but also by how much these methods make desired safety properties more correlated with scale." This is potentially a dangerous instance of the availability heuristic. If some safety properties will not correlate with scale, then does that mean they are less important to solve? How do we know that in changing a safety property measurement to improve with scale we have not changed its essential character so that it no longer measures what we want it to measure? An adequate answer to this question is my most important commentary.
* The prior comment also seems to contrast with: "We argue researcher time for developing methods should be allocated toward solving benchmarks that won’t be solved with scale and general capabilities advancements." Are we not trying to produce safety benchmarks tightly coupled to scaling?
* Strange: "...non-capability properties of networks, such as safety attributes."

**Strengths:**

There are significant methodological issues facing the LLM safety community that are expressed at the intersection of safety and capability/scale. This paper makes a strong advancement in the direction of examining this issue.

**Additional Feedback:**

"...non-capability properties of networks, such as safety attributes." Capability and safety attributes are intimately linked. It is strange seeing them so separated as this text.

"...success in new safety methods should be measured not only by improvements in safety benchmark scores, but also by how much these methods make desired safety properties more correlated with scale." This seems reasonable, but it is perhaps more instructive of the nature of the dataset being trained against. In reinforcement learning, sparse reward signal long challenged the state of the art. It was not an increase in scale that solved the problem, but it was a change in the reward signal that then restored performance's correlation with scale. A meta-analysis of where the data may be lacking for certain benchmarks would be an additionally valuable contribution, as well as perhaps looking at whether the training objective linked to that data may not be adequate.

**Correctness:**

The recommendations made by the paper are not necessarily well supported and require additional evidence or prose supporting the argument. I might be convinced of correctness, but I am not presently convinced that these are the right recommendations to make to the research community.

**Documentation:**

No dataset.

**Ethics:**

No ethical issues.

**Limitations:**

Yes, the authors do an adequate job of addressing the limitations of the work.

No negative societal impacts of this work are implicated.

**Opportunities For Improvement:**

The authors show incomplete engagement with the scale literature. In particular, in the model compression literature "scale" may be viewed as a shortcut to capabilities rather than a key (e.g., papers on pruning as a method of hyperparameter search, the lottery ticket hypothesis). That critique so far is playing out well with industry movements, where models of great scale are first placed on the market, then greatly optimized versions of the model are deployed at the same capability level but with a fraction of the model scale.

This is more a critique of the scaling hypothesis as it relates to this work than to this work itself. Appealing to scale appears to be unnecessary for the main results of the paper which is an interesting analysis of the nature of safety as it relates to the variety of capabilities exhibited by the model. In short, the recommendations for where the research world should focus, "We argue researcher time for developing methods should be allocated toward solving benchmarks that won’t be solved with scale and general capabilities advancements" is a great one when it comes to the second part (general capabilities), but not the first (scale). In effect, the paper would then be arguing for finding exclusively those modeling elements required for exhibiting a singular safety property and none of the non-essential properties.

**Relation To Prior Work:**

Additional discussion of meta-benchmarks would be valuable.

**Summary And Contributions:**

* The authors construct a composite benchmark of many benchmarks meant to measure the "capabilities component" of the benchmarked system.
* By accounting for the correlations between the component benchmarks, a measure asserted to be strongly associated with model "capability" is produced.
* The composite measure correlates strongly with model scale
* The composite measure is then compared to a variety of safety related benchmarks to find how capabilities influence measured safety properties
* A variety of research prioritization and practice assertions are made, including the prioritization of safety research that is independent of properties of scale.

---

> ### Author Rebuttal · Authors · 2024-08-15
>
> Thank you for your detailed analysis of our paper. We hope the following response addresses your concerns.
>
> **Safety properties that do not correlate with general capabilities are more important**
>
> > If some safety properties will not correlate with scale, then does that mean they are less important to solve?
>
> In contrast, we argue that safety properties that do not correlate with general capabilities (and thus are not solved by future more capable models) are more important to solve.
>
> The point we intended to make – which may have been unclear in our writing – was that if a given safety metric is uncorrelated with general capabilities, a productive research direction is to develop new *methods* that increase its capabilities correlation. For example, [Howe et al., 2024] find that the adversarial robustness of LLMs does not improve with scale by default, but does improve with scale after incorporating adversarial training. Once a high capabilities correlation is obtained, then researchers can rely on progress in general capabilities to further improve the safety metric (assuming that such progress is sufficient to solve the task; otherwise, see “Clarifying false sense of security” in our reply to Reviewer VKj4).
>
> Insofar as it is not tractable to tie these safety properties to general capabilities, that should be a further indication of the difficulty and importance of that specific safety problem, rather than an indication not to work on that problem.
>
> We believe the previous writing – while intending to make the points above – was confusing, and we will change our discussion to more clearly reflect the points made here. We hope this makes more sense and we thank you for your feedback.
>
> [Howe et al., 2024]: “Exploring Scaling Trends in LLM Robustness”. Howe et al. arXiv 2024.
>
> **We do not advocate for new safety benchmarks correlated with scale**
>
> > How do we know that in changing a safety property measurement to improve with scale we have not changed its essential character so that it no longer measures what we want it to measure? An adequate answer to this question is my most important commentary.
>
> We agree that keeping safety benchmarks fixed is important. We do not ask that researchers alter the benchmarks they use to make them more correlated with capabilities. Instead, we argue that safety methods (e.g. fine-tuning, refusal training, robustness training) may improve correlations.
>
> **Clarifying recommendations**
>
> > The prior comment also seems to contrast with: "We argue researcher time for developing methods should be allocated toward solving benchmarks that won’t be solved with scale and general capabilities advancements." Are we not trying to produce safety benchmarks tightly coupled to scaling?
>
> We don’t intend to make the point that benchmarks should be designed to have high correlations. Rather, our two suggestions occur in two different contexts, which are that:
> 1. For benchmark selection (and broadly, research problem selection), researchers should allocate more time toward creating and climbing benchmarks with low correlation and thus represent problems that will not be solved with general capabilities advancements using current methods.
> 2. For technique development, a fruitful research direction is to develop new *safety methods* that increase the capabilities correlation, ensuring that the safety benchmarks will improve in the future as general capabilities improve.
>
> So selected safety benchmarks and research problems should be less correlated, while safety techniques should aim to increase correlation.
>
> We think this point was understandably unclear in the introduction and discussion, and drove doubts about the recommendations made in your review. In our future revision, we will significantly clarify what our two separate recommendations are to remove confusion.
>
> **Acknowledging the distinction between capabilities and scale**
>
> We fully agree that scale is not the only factor in determining general capabilities. You make a great point about how researchers have made advances in capabilities without necessarily increasing scale.
>
> Across the models that we study, we find that scale is highly correlated with general capabilities, but several open-weight models have recently been released that increase general capabilities without increasing model size (e.g., Llama 3.1). We plan to incorporate these into the updated experiments, as well as a selection of distilled models. We will also add a brief discussion of the different factors that can contribute to general capabilities, including scale, data, and algorithms. Finally, we will update the text in the paper to decouple scale from general capabilities – especially because our method is specific to capabilities benchmarks rather than scale. Thank you for your suggestion.
>
> **Other**
>
> We will correct the phrase “non-capability properties of networks, such as safety attributes” (and similarly unclear phrases). Instead, we will make it clear what we refer to: “safety properties of models” and “upstream/latent general capabilities of models.”
>
> If we have addressed your concerns, we kindly ask that you consider raising your score.

---

> > ### Author Response · Authors · 2024-08-23
> > **Friendly reminder**
> >
> > This is just a friendly reminder that the discussion period is ending soon. If you have any followup questions about the paper or our reply to your initial concerns, we would be glad to answer them.

---

> > > ### Comment · Reviewer_iAJj · 2024-08-26
> > > **Increasing score**
> > >
> > > I am increasing my score on the basis of the author rebuttal. I may increase it another level following additional thought and/or discussion.

---

> > > > ### Author Rebuttal · Authors · 2024-09-01
> > > >
> > > > Thank you for increasing your score.
> > > >
> > > > **Experimental results: Elucidating the distinction between capabilities and scale**
> > > >
> > > > We have finished running some of the experiments mentioned in our reply (the Llama 3.1 8B and 70B experiments), the results of which are shown in the PDF attached.*
> > > >
> > > > These results indicate that model size is indeed not the only factor in determining general capabilities and demonstrates that our analysis can account for these additional factors. We plan to update the text of the paper to primarily allude to model capabilities – rather than scale, training FLOP, or model size – especially because you thoughtfully pointed out that various distilled models or other training methods may further decouple capabilities from scale.
> > > >
> > > > *Note: The capabilities component was slightly modified/recalculated to remove a few missing capabilities evaluations that are still running on the Llama 3.1 models. Therefore, capabilities scores may change slightly in our final paper.
> > > >
> > > > **Other suggestions**
> > > >
> > > > You make an excellent point that rather than solely focusing on the development of techniques to correlate safety properties with scale, our meta-analysis may also elucidate where data or training objectives may be inadequate. While a meta-analysis of where training data is lacking may be beyond the scope of our research paper, we will add your thoughtful points to our paper discussion as potential future work.
> > > >
> > > > We also plan to add additional discussion on meta-benchmarks and previous criteria put forward for benchmarks. We agree that it would be helpful for the reader to understand previous work and perspectives on evaluating benchmarks, which will help frame our new recommendations for future ML safety benchmarks.

---

### Official Review · Reviewer_VKj4 · 2024-07-27
**Review for "Quantifying the Bitter Lesson: How Safety Benchmarks Measure Capabilities Instead of Safety"**

**Rating:** 7
**Confidence:** 4
**Clarity:** Yes

**Review:**

- The paper demonstrates a high level of methodological rigor by employing spectral analysis to understand the relationship between safety benchmarks and general model capabilities.
- Offers a fresh perspective by quantitatively analyzing the correlation between safety benchmarks and general capabilities, an area that has not been extensively explored.
- The discussion section is on point - the authors suggest focusing on benchmarks with low capabilities correlation to ensure dedicated safety efforts. Measuring properties that scale can still predict future risks and guide safety advancements. New evaluations should report their correlation with capabilities to validate their relevance. Overall, the paper calls for a balanced approach to aligning safety metrics with capabilities.


### Pros
- The use of spectral analysis and principal component analysis provides a framework for understanding the data.
- The study covers a wide range of benchmarks and models, ensuring that the findings are broadly applicable
- Strong discussion section with practical recommendations


### Cons
- Authors calling existing work on designing benchmarks as wasteful, assuming that primary problem with benchmarks is correlation with capability, undermining other benefits and problems.
- Potential for misinterpretation of the research where the belief that general capabilities improve safety can create a false sense of security.  If researchers believe that scaling up models will automatically solve safety issues, there may be less incentive to develop targeted safety interventions and methods.

**Strengths:**

The findings are highly relevant to the AI safety community, which is tasked with ensuring that AI systems are safe and reliable. The insights and methodologies proposed can guide researchers in creating more effective and meaningful safety benchmarks, impacting a wide range of AI applications. The paper can influence future studies and resource allocation in AI safety A LOT, and if this research is reproduced by companies on their benchmarks, they can prioritize which classes of harm they need to solve by safety techniques vs the ones that will either positively or negatively get impacted with scale.

**Additional Feedback:**

Provide a more detailed explanation of the spectral analysis methodology, including any assumptions made and the rationale behind using this approach. Maybe a step-by-step descriptions could help clarify this for readers who are less familiar with the technique.

**Correctness:**

Yes, overall the methodology is sound but authors can solidify by
- Use methods like kernel-based correlations and causal analysis to capture more complex relationships between capabilities and safety metrics.
- Provide discussion on all the assumptions used in spectral analysis

**Documentation:**

Authors should provide code to simplify the process of reproducibility.

**Limitations:**

- The authors should explore a wider range of limitations beyond the correlation between safety and capabilities. This could include limitations related to the diversity of data, potential biases in benchmark design, and the generalizability of their findings across different types of AI models
- Authors should provide explanation of assumptions in their spectral analysis. This assumes that statistical properties of the data do not change over time or with different samples. The authors need to examine if this assumption is reasonable in the context of evolving AI models and benchmarks.
- Conduct causal analysis to determine if improvements in capabilities directly cause improvements in safety metrics, rather than just being correlated
- Perform longitudinal studies to assess how the correlation between capabilities and safety metrics evolves over time as models are scaled and improved.
- Explore non linear methods to capture correlation between capability and safety. If the relationship between a model's size and safety shows that beyond a certain size, further increases in model size lead to diminishing improvements in safety, a non-linear method like kernel correlation could effectively capture this plateau effect, whereas linear methods might not.

**Opportunities For Improvement:**

Narrow Scope of Solutions -  While the paper identifies an important issue, the proposed solutions are somewhat general. They call for better benchmarks and more focused research but do not provide detailed methodologies or specific examples of such benchmarks. This might reduce the paper's immediate impact on ongoing research and benchmark development.

**Relation To Prior Work:**

Yes

**Summary And Contributions:**

The paper addresses the issue that many safety benchmarks in AI research are highly correlated with general capabilities benchmarks. This correlation suggests that improvements in these benchmarks often result from scaling up model size and compute rather than addressing specific safety concerns. Authors leverage spectral analysis to measure an underlying capabilities component, which explains most variation in model performance. They analyze existing safety benchmarks, and find that variance in model performance on many safety benchmarks is largely explained by the capabilities component.


### Contributions
- Demonstrates that performance on many safety benchmarks is largely explained by an underlying capabilities component, indicating a strong correlation with general model capabilities.
- Safety Researchers/companies can do similar analysis of their benchmarks and categorize safety tasks into three classes based on their correlation with model scale: those that improve with scale, those that improve with tuning, and those unsolved by scaling, so that they can focus on safety-specific improvement.
- Suggests that future safety benchmarks should report their correlation with capabilities to ensure they measure meaningful safety properties, and that researcher time for developing methods should be allocated toward solving benchmarks that won’t be solved with scale and general capabilities advancements. This ensures that safety improvements are due to specific safety measures rather than general capability enhancements.
- The findings suggest a need for better-designed safety benchmarks that can isolate and measure safety-specific attributes. This can lead to more focused and effective safety research, ultimately resulting in safer AI systems.

---

> ### Author Rebuttal · Authors · 2024-08-15
>
> Thank you for your detailed analysis of our paper. We appreciated your comment on how our work could influence resource allocation and safety – this is exactly what we hope to do. We hope the following response addresses your concerns.
>
> **Removing the “wasteful” claim**
>
> We agree that having a high capabilities correlation does not inherently mean that a safety benchmark is wasteful. In fact, we clarify in the discussion that “evaluations showing strong correlation or anti-correlation with capabilities are valuable for tracking the evolution of dangerous capabilities and ensuring precise measurement of safety metrics, even if they are eventually solved with scale”. Because of this, we agree that the “wasteful” questioning should be removed from the abstract for clarity and have updated the paper accordingly thanks to your suggestion.
>
> **Clarifying false sense of security**
>
> In our paper, we argue that safety techniques can also tie safety properties of interest with capabilities. For example, previous work has found that SFT, DPO, and RLHF on harmlessness data (refusal training) may increase the correlation between refusal (a safety property) and capabilities.
>
> However, we agree that wholly relying on capabilities correlations can provide a false sense of security. In particular, there are some cases in our evaluations where tasks with high capabilities correlations will be improved but not completely solved as general capabilities improve. We will add this point to the discussion section in the updated paper. Thank you for your suggestion.
>
> **Wider range of limitations**
>
> In our paper, we focus on the limitations associated with having a high capabilities correlation, but we agree that safety benchmarks face other limitations as well. Thanks to your suggestion, we will expand our discussion to highlight other potential problems with the design of safety benchmarks, including data diversity and biases that can affect the generalizability of results. For example, we can mention how various models have underperformed on GSM1K relative to GSM8K, and explain how this may influence correlations and results.
>
> **Assumptions in spectral analysis**
>
> We agree that capabilities correlations can shift over time as new models are introduced. In several experiments, we show how advances in instruction-tuning have already altered capabilities correlations. Thus, our results are not static, but rather a snapshot of capabilities correlations across models introduced in the past few years. We will clarify this point in the updated paper.
>
> **Non-linear methods**
>
> While standard correlation captures first-order effects, we agree that methods like kernel correlation would provide additional insight. We will investigate adding an analysis with kernel correlation to the updated paper. Thank you for your suggestion.
>
> **Causal analysis**
>
> While doing direct causal analysis would require training many models, we can run a natural experiment using model families – an analysis where we know various models have been trained the same way (e.g. same dataset and instruction fine-tuning method), with the only difference being scale. This can be done, for example, on the Llama or Qwen family of models. We will conduct this experiment and report the results.

---

### Decision · Program_Chairs · 2024-09-26

**Decision:**

Accept (Poster)

**Comment:**

This paper investigates the relationship between LLM performance on safety benchmarks and general LLM capabilities, in the spirit of the Bitter Lesson. The authors find that improvements on many safety benchmarks are largely explained by improvements in model capabilities. Consequently, the authors argue that there is a need for novel safety benchmarks that do not correlate strongly with scale/capabilities.

Reviewer opinions on this paper all lean positive, with final scores of 7, 6, 6, and 7. The quality of all four reviews is high, with each reviewer providing extensive feedback on different aspects of the paper. The authors engaged with this feedback very thoroughly in their author responses, and most reviewers increased their scores accordingly.

The main reviewer criticisms revolve around clarity and framing of the paper. I believe these criticisms to be warranted. For example, reviewer iAJj makes a good point around disentangling claims about scale from claims about capabilities. Reviewer 6eSn’s points around defining safety also resonated with me.

Another concern I would add is that this paper, in my opinion, lacks conceptual engagement with how safety and capabilities relate to each other. The empirical evidence is interesting and comprehensive, but there is little attempt to explain the results, i.e. why certain safety benchmarks correlate with scale/capabilities while others do not. Doing so would also help with deriving more actionable recommendations around what concrete kind of safety benchmarks *will* continue to be useful.

Overall, in line with reviewer scores and the scope of the suggested/planned revisions, I am recommending acceptance of the paper for poster presentation.